# Template-Free Hydrothermal Synthesis of Octahedron-, Diamond-, and Plate-like ZrO_2_ Mono-Dispersions

**DOI:** 10.3390/nano12193405

**Published:** 2022-09-28

**Authors:** Ling Gao, Hao Zhi, Shengnan Zhang, Shifeng Liu

**Affiliations:** 1School of Materials Science and Chemical Engineering, Xi’an Technological University, Xi’an 710021, China; 2College of Metallurgical Engineering, Xi’an University of Architecture and Technology, Xi’an 710055, China; 3Northwest Institute for Nonferrous Metal Research, Xi’an 710016, China

**Keywords:** hydrothermal, zirconia, morphology, NaBF_4_

## Abstract

Anisotropic ZrO${}_{2}$ particles with octahedron-, diamond- and plate-like morphologies are successfully synthesized by a facile hydrothermal treatment approach using NaBF${}_{4}$ as mineralizer. The concentration of mineralizers play a crucial role on the formation of shape-controlled ZrO${}_{2}$ particles thus affect the particle size. With the increasing concentration of mineralizer, the crystalline sizes of the primary single-crystal and the secondary particle size both increase. With the introduction of NaBF${}_{4}$, F${}^{-}$ plays an essential role in tuning the crystallinity and size of primary ZrO${}_{2}$ nanorods along [001] direction. The synergistic effect of F${}^{-}$ and B${}^{3+}$ result in different epitaxial growth rate. And the secondary particles mainly crystallize on the small primary nanoparticles through the oriented attachment mechanism. The as-prepared ZrO${}_{2}$ particles with different sizes and shapes exhibit different photocatalytic efficiency for the degradation of organic dyes. Under UV irradiation, the highest MB degradation rate of 88% was observed within 60 min for ZrO${}_{2}$ photocatalyst synthesized with 0.01 mol/L NaBF${}_{4}$ mineralizer.

## 1. Introduction

In recent years, controlled preparation of nanocrystals with specific sizes and shapes have been extensively investigated in studies involving the synthesis of nanoparticles and searching for suitable methods for growing anisotropic crystals. Anisotropic shaped zirconia structures such as nano-wires, nano-belts or platelets are thought to be useful starting materials for the oriented growth of zirconia ceramics [1] and fabrication of shape-dependent zirconia catalyst or catalytic support [2], luminescent materials [3], biological materials [4], gate dielectric in metaloxide semiconductor (MOS) devices [5] and solid state oxide fuel cell [6]. In the past few years, studies on different wet-chemical synthesize route are developed to synthesize ZrO${}_{2}$ with one-dimensional (1D) morphologies such as nanowire [7], nanobelts [8], nanotube [9], nanorods [10,11]. In stark contrast, investigations of 2D (except thin film) zirconia nano- or micro-particles remain unexploited, largely because of the formidable challenges associated with the controlled dimensions and phase structure. The most common and available synthesis of flake zirconia particles is the hydrothermal synthesis with zirconium salts and sulfuric acid as starting materials. But these synthesis routes involve long time treatment and multiple steps [12,13,14]. Moreover, all the synthesis approaches such as sol-gel [15,16], ionic-liquid route [17], self-assemble route [18], hydrothermal synthesis [19] and molten salt method [20] employed during the preparation of anistropic nanoparticles require an organic additives and a calcination treatment to induce crystallization which lead to the agglomeration of nanoparticles.

In comparison, hydrothermal synthesis is an ideal technique for synthesizing materials which can effectively control the crystal growth process and prevent hard agglomeration. Classically, growth of crystals has been thought to occur by atom-by-atom addition to an inorganic or organic template or by dissolution of unstable phases and reprecipitation of the more stable phase. In addition to the ion-mediated classical crystal growth, particles can grow by aggregation with other particles involving a mesoscopic transformation process. After the particles grow to a stable size, they will grow by combining with smaller unstable nuclei rather than by collisions with other stable particles [21]. In most of the hydrothermal routes, alkali metal hydroxides mineralizer and hydrous zirconia precursor resulting from hydrolysis of soluble zirconium salt with alkaline base are introduced to prepare ZrO${}_{2}$ particles. However, spheroidal particles are mostly obtained in this way, which restricts the application in many field. Organic additives have been found to be the key factor for anisotropic growth, and many surfactants have been used to synthesize anisotropic nanomaterials. However, Weller provided that perfect ZnO nanorods can be conveniently self-assembled from small quasi-spherical nanoparticles without any organic additives based on the oriented attachment mechanism [22]. Teng pointed out that the hydrothermal treatment of titanate nanotube suspensions under an acidic environment without any organic additives could result in the formation of single crystalline anatase nanorods. Preparation of ZrO${}_{2}$ anisotropic materials from solutions absent of any organic surfactants or templates has rarely been reported.

Here, we introduced a facile and repeatable approach based upon a hydrothermal method for growing well-defined multi-morphology monoclinic ZrO${}_{2}$ sub-micron particles, which could be easily achieved by introducing NaBF${}_{4}$ with variation concentration as mineralizer. The crystal structure, microstructure and the photocatalytic property were investigated for the as-synthesised particles. To the best of our knowledge, these sub-micro materials provided the first evidence of ZrO${}_{2}$ crystals with octahedron-, diamond- and plate-like morphologies by simply adjusting the concentration of mineralizer.

## 2. Experiment

In a typical synthsis of the ZrO${}_{2}$ particles, ZrOCl${}_{2}$·8H${}_{2}$O (SCRC, analytical reagent, AR) and NaBF${}_{4}$ (SCRC, AR) were used without further purification. A homogeneous solution with ZrOCl${}_{2}$·8H${}_{2}$O concentration of 0.1 mol/L and NaBF${}_{4}$ concentration of 0∼0.05 mol/L was formed by the appropriate amount of material dissolved in 30 mL deionized water. Then the mixed solution was transferred into a 50 mL Teflon-lined hydrothermal reactor and heated at $200{\;}^{\circ }C$ for 12 h. Upon cooling the resulting precipitate was filtered and washed with distilled water for several times. And finally, the precipitate was dried at $100{\;}^{\circ }C$ for 24 h. In order to describe simply, the samples synthesized with different concentration of NaBF${}_{4}$ as mineralizer are denoted as HT-ZrO${}_{2}$(*x*), where *x* represents the molar concentration of NaBF${}_{4}$.

The as-synthesized products were characterized by X-ray diffractometer (Rigaku D/max-2550, Japan) with Cu-Kα radiation to analyze the phases composition. Scanning electron microscope (Zeiss JEM-6460, GemimiSEM 560, Zeiss, Germany) and TEM (JEOL JEM-2100Plus, JEOL, Japan) were performed to give a direct view of the morphology of the products. Samples for electron microscopy were prepared by air-drying a drop of a sonicated suspension of the dried precipitation in ethanol onto carbon-coated copper electron microscope grids. High resolution TEM (HRTEM) and selected area electron diffraction (SAED) were performed with a CM200FEG microscope operating at 200 kV and equipped with a field-emission gun. X-ray photoelectron spectroscopy (XPS) was carried out on an ESCALAB 250Xi high-performance electron spectrometer (Thermo Fisher Scientific Inc., Waltham, USA), using monochromatic Al K-alpha radiation. UV-vis absorption measurement was carried out in a UV-vis spectrophotometer (UV-2100S, Shimadzu, Kyoto, Japan).

The photocatalytic efficiency of the samples was evaluated by the degradation of methylene blue (MB) under the irradiation of a 500-W halogen lamp (Philips). The light intensity is around 4.5 mW/cm${}^{2}$. The photocatalysts (20 mg) were soaked in 50 mL of 20 mg/L MB solution and kept in the dark for 30 min to establish an adsorption–desorption equilibrium. The light was then turned on to start the reaction. The change in the MB concentration of each degraded solution was monitored on a UV-vis spectrophotometer (JASCO V-530, Japan). The photocatalytic efficiency was calculated according to the following equation: D = (C${}_{0}$ − C)/C${}_{0}$ × 100%, where D present decolorization efficiency, C${}_{0}$ and C represent the initial concentration of MB before irradiation and the concentration of MB after irradiation for a given time, respectively.

## 3. Results and Discussion

Figure 1 presents the X-ray diffraction patterns of the as-synthesized powders with different concentration of NaBF${}_{4}$ (0∼0.05 mol/L). It can be observed that monoclinic ZrO${}_{2}$ phase can be formed with the NaBF${}_{4}$ concentration lower than 0.05 mol/L. But when the concentration of mineralizer reaches 0.05 mol/L, diffraction peaks of secondary phase can be obviously detected, which suggests that higher mineralized concentration is not suitable for the preparation of single phased ZrO${}_{2}$ particles. It is a pity that the diffraction peaks of secondary phase can not be indexed by any patterns based on the existing PDF data. With further study of the materials synthesized with higer NaBF${}_{4}$ concentration (>0.5 mol/L), it is found that Na${}_{2}$ZrF${}_{6}$ was formed. Though the diffraction peaks do not fit the Na${}_{2}$ZrF${}_{6}$ (No. 49-0108) perfectly, we confirm that the new phase should be a king of fluoro-zirconium compound. The detail information about the synthesized products with higher NaBF${}_{4}$ concentration will be discussed in another article. The inset figure shows the amplifying XRD patterns of 2θ range from 26${}^{\circ }$ to 37${}^{\circ }$. It clearly shows that as the mineralizer concentration increases, the diffraction intensity and crystallization of monoclinic ZrO${}_{2}$ both increase gradually, with no significant shift of diffraction peak position. The apparent crystallite size of all the specimens determined by Scherrer’s equation are given in Table 1. Consequently, it is found that the apparent crystallite size of ZrO${}_{2}$ increases from 12 to 29 nm as the NaBF${}_{4}$ concentration increases from 0 to 0.03 mol/L.

Figure 2a–f shows the SEM images of ZrO${}_{2}$ particles synthesized with and without NaBF${}_{4}$ mineralizer. The particle shape and size are found to be largely dependent on the mineralizer concentration. Under the same hydrothermal condition, the particle size increases with increasing concentration of NaBF${}_{4}$. The detail information of the average secondary particles sizes determined form the SEM photographs are also shown in Table 1. The secondary particle size of ZrO${}_{2}$ synthesized from ZrOCl${}_{2}$ solution with and without NaBF${}_{4}$ increases form 50 nm to 2 µm as the NaBF${}_{4}$ concentration increasing from 0 to 0.03 mol/L. More over, it can be clearly observed that all the powders are composed of nano-crystals clusters, and the morphologies change significantly with different mineralizer concentration. The secondary particles synthesized without mineralizer exhibit irregular shape. By increasing the mineralizer concentration, the particle shapes change from column-like to diamond-like, and finally to plate-like shapes. Further more, columnar particles are composed with rod-like crystal, diamond- and plate-like particles have layered structure. When the concentration reaches 0.05 mol/L, plate-like particles are easily crushed, which is due to the formation of the new phase. Cross growth particles can be seen all over the samples.

Subsequently, to clarify the detail structure information of the secondary particles with various shapes, TEM measurement are performed on various shaped ZrO${}_{2}$ particles. Figure 3a–e shows the TEM photographs of the ZrO${}_{2}$ particles synthesized with and without NaBF${}_{4}$. It can be clearly seen that the secondary particles synthesized with low mineralizer concentration (Figure 3a–c) are formed with primary rod-like nanocrystalline in a particular way. There are two kinds of epitaxy patterns: one is epitaxy along with the longitudinal axis of nanorod, the other is epitaxy parallel to the longitudinal axis of nanorod. And two groups of epitaxy crystallines in pattern one present a regular non-perpendicularly crossing distribution. Because of the different epitaxy growth rate of the two patterns, the particles grow into a structure with thicker center region and thinner edges. With the addition of NaBF${}_{4}$, the growth rate of particles in the first pattern is faster than that of the second one, which results in the replacement of short columnar particles (Figure 3a) to long columnar particles (Figure 3b,c). According to the characteristics of particles with thick center and thin edge, we define them as octahedron-like particles. When increasing the NaBF${}_{4}$ concentration to 0.02 mol/L, the thicknesses of both the center and edge regions become uniform, but the particle length is still longer than the width and thickness. Further increasing the mineralizer concentration, oval plate-like particles with micro order size are produced.

The surface chemical composition and chemical states of the HT-ZrO${}_{2}$ (0.02) samples are investigated using XPS analysis, the survey spectra and the high resolution spectra for Zr 3d, F 1s and O 1s are shown in Figure 4. The survey XPS spectra reveal that the selected samples are composed of Zr, O and F (Figure 4a). Zr 3d spectra consist of Zr 3d${}_{5/2}$ at 182.2 eV and Zr 3d${}_{3/2}$ at 184.6 eV peaks with a separation of 2.4 eV, indicating the presence of fully oxidized state of Zr${}^{4+}$ in ZrO${}_{2}$ [23]. In Figure 4c, the signals for F 1s are obtained at 685.0 eV and 687.1 eV. The main contribution is assigned to F${}^{-}$ ions physically adsorbed on the surface of ZrO${}_{2}$ [24]. The minor contribution of the F in ZrO${}_{2}$, which is probably formed by nucleophilic substitution reaction of F${}^{-}$ ions and zirconium hydroxide during the hydrothermal process [25]. The O 1s signal can be well fitted into two peaks. The two separate peaks located at 530.3 eV and 531.7 eV can be attributed to Zr–O–Zr (lattice oxygen) and Zr–OH, respectively [26].

In order to identify the formation of the secondary particles, the high resolution TEM images of octahedron-like particle are shown in Figure 5. Figure 5a,b show the atomically flat edge with a spacing of 0.505 nm between adjacent lattice planes that corresponds to the distance of two (100) crystal planes, proving (010) to be the exposed surface. HRTEM images of the same position on [110] zone axis are shown in Figure 4c,d. High resolution images allow the observation that the rod-like ZrO${}_{2}$ tend to grow along [001] direction, as indicated in the figure. The primary ZrO${}_{2}$ nanobar is regarded as growth union with the oriented attachment growth mechanism. (100) plane is the symmetry and twinning plane of monoclinic ZrO${}_{2}$, so that the two nanorods use of their (100) crystal planes to form a grain attachment. When the two freestanding nanobars join together along the (100) plane to form a oriented aggregation, the c direction of the crystal axis crosses each other, and then other nanobars will oriented attach on the primary two nanobars. The naturally grown crystal planes for the monoclinic ZrO${}_{2}$ nanorod determined to be (001), (100) and (010), and the growth rate of the ZrO${}_{2}$ nanobar have the following hierarchical order of (001)>(100)>(010) [27]. It is interesting to find that all the three principal crystal planes are able to participate in the observed aggregative attachment. For example, as depicted in Figure 4, the inter-growth rate of c(001) plane is the highest, when nanobars oriented attached along this plane, a crossing frame is first formed. At the same time, an extension along the (100) plane, in addition to the longest dimention in the [001], would give rise to an overall two-dimensional attachment. Furthermore, a stacking of these nanobars along the (010) would add another dimension to the crystal aggregates, which virtually produces a three-dimensional octahedron-like structure, although overall dimensional lengths still keep the same hierarchical order as that of the crystal plane attaching rate.

With the increasing concentration, due to the formation mechanism of diamond-, and plate-like ZrO${}_{2}$ particles are similar, the HRTEM images on the same zone axis at different positions of the diamond-like particles are shown in Figure 6. It is interesting to note that, the particles are not stacked with nanobar but nanosheet. It is a pity that the shape of the nanosheet can not be clearly observed because of its big particle size and thickness. The interplanar spacings of 0.263 nm and 0.3701 nm determined from Figure 6a–c correspond to the (002) and (011) crystal planes, proving (100) to be the exposed surface. The (100) planes possess a strong capacity for crystal inter-overlapping. We predict the formation of diamond- and plate-like particles as follow: Firstly, nanosheets use their (001) and (010) planes to form a coplanar sheet-like structure, due to the attach rate of (001) plane is faster than the one of the (010) plane. So the length of the particle along [001] direction is longer than the width along [010] direction. Then the coplanar sheet-like structure is coupled with another crystal underneath and upward the oriented attachment of their (100) planes, and finally forms a stratified structure.

With the XPS result, it is found that B${}^{3+}$ ions can not be detected on the ZrO${}_{2}$ particle surface. Therefore, it can be deduced that F${}^{-}$ plays an essential role in tuning the crystallinity and size of primary ZrO${}_{2}$ nanorods and nanosheet. F${}^{-}$ has been widely used in the field of titanium dioxide synthesis instead of OH${}^{-}$ as the mineralizer, structure-directing agent or morphology control agent [28]. As the mineralizer, it can significantly promote the crystallization process of TiO${}_{2}$. The same function is also found in our case, and the following reasons are proposed to explain the effects of F${}^{-}$ on the crystallization process of ZrO${}_{2}$ nanorods/nanosheets. Firstly, the existence of F${}^{-}$ increases the chemical potential of the solution, which is favorable for the growth of nanostructure. Moreover, it can significantly promote the solubility of zirconium precursor and decrease the viscosity of the solution, which facilitates the mobility and diffusion of the components in the hydrothermal system and allows atoms, ions, and molecules to adopt the appropriate position in developing crystal lattices [29]. B${}^{3+}$ ions act as morphology regulator to make the primary ZrO${}_{2}$ crystal oriented attach to form multi-morphology secondary particle. B${}^{3+}$ ions in the solution modify the Lewis acid sites of the ZrO${}_{2}$ surface. Zhou found that ZrO${}_{2}$ doped with B had more Lewis acid sites, and postulated that the Lewis acid sites were located at the Zr atoms, and the electron-deficient B${}^{3+}$ could modify the Lewis acidity of Zr via an inductive effect [30]. A similar effect of B on the Lewis acidity of Zr${}^{4+}$ is expected in this study. ZrO${}_{2}$ with more Lewis acid sites have a strong attraction to other lone pair electrons, such as F${}^{-}$ and OH${}^{-}$ adsorbed on the surface of another ZrO${}_{2}$ crystal. When two nanoparticles approach each other close enough, they are mutually attracted by van der Waals forces. The adjacent nanoparticles are self-assembled by sharing a common crystallographic orientation and docking of these particles at a planar interface. Thus bigger particles can grow from small primary nanoparticles through an oriented attachment mechanism.

In the course of experiment, it is found that the pH values of the initial solution and that of the solution after the reaction were all less than 2. Under this condition, the possible chemical reactions in the hydrothermal process are as follow: firstly, ZrOCl${}_{2}$ undergoes hydrolysis reaction to form zirconium hydroxide (Equation (Equation 1)) [31], NaBF${}_{4}$ hydrolyzes by heating to form hydrofluoric acid (Equation (Equation 2)); secondly, the hydroxyl (-OH) groups on the surface of Zr(OH)${}_{4}$ has a ligand exchange with the F${}^{-}$ ions in water solution, and Zr-OH is replaced by Zr-F (Equation (Equation 3)); finally, as the surface hydroxyl groups are further replaced by F ions, the Zr atoms on the surface of the crystal will eventually dissolve into the aqueous solution as Zr${}^{IV}$ complexes (such as ZrF${}_{6}$${}^{2-}$), which lead to the dissolution of the insoluble Zr(OH)${}_{4}$ in water [32].
(1)$${\mathrm{ZrOCl}}_{2}+3H_{2}O\;\;\to \;\;{\mathrm{Zr}(\mathrm{OH})}_{4}+2H^{+}+2{\mathrm{Cl}}^{-}$$
(2)$$\mathrm{BF}{{}_{4}}^{-}+H_{2}O\;\;\to \;\;{\mathrm{BF}}_{3}{\mathrm{OH}}^{-}+\mathrm{HF}$$
(3)$$\mathrm{Zr}-\mathrm{OH}+H^{+}+F^{-}\;\;\to \;\;\equiv \mathrm{Zr}-F+H_{2}O$$

Figure 7a displays the UV-vis diffuser reflectance of the HT-ZrO${}_{2}$(*x*) (*x* = 0, 0.01 and 0.03). The optical absorbtion spectras of the ZrO${}_{2}$ synthesized with and without mineralizer all show a strong absorbtion with the wavelength shorter than 254 nm and a light absorbtion with the wavelength between 250 to 400 nm. It means that the increasing particle size and morphology could not change the optical absorbtion property. The Wood and Tauc’s polt for determining band gap energy is shown in the insert picture. The direct band gap for all the samples are all the same with 4.94 eV, and the band gap introduced by defect increases with the addition of NaBF${}_{4}$ mineralizer. Figure 7b shows the decolorization efficiency of MB under simulated solar irradiation with in 60 min for various ZrO${}_{2}$ particles. The decolorization efficiency of MB increases with octahedron-like particles, but when the particle shape changes into diamond- or plate-like morphologies, the decolorization efficiency decreases. Especially for the plate-like particle, the decolorization efficiency is lower than the ZrO${}_{2}$ samples synthesized without mineralizer, it may be due to the enlarged particle size with micrometer. For the octahedron-like particles synthesized with the concentration of NaBF${}_{4}$ 0.01 mol/L, the maximum decolorization efficiency of MB reach 88%.

## 4. Conclusions

In summary, epitaxial ZrO${}_{2}$ submicrometer particles with various morpholoyies were synthesized using a facile hydrothermal method at $200{\;}^{\circ }$C for 12 h by adjusting the concentration of NaBF${}_{4}$. The experimental results indicated that the concentration of mineralizers played the crucial role in the formation of shape-controlled ZrO${}_{2}$ particles and its particle size. Based on the micro-structure analysis, epitaxy of monocilnic ZrO${}_{2}$ nanorod crystalline synthesized with low NaBF${}_{4}$ concentration (<0.01 mol/L) resulted in the formation of octahedral-like particles. With increasing concentration of NaBF${}_{4}$ (up to 0.03 mol/L), diamond-like and plate-like ZrO${}_{2}$ particles were formed, and the particle size increased from sub-micrometer to micrometer-grade. The synergistic effect of F${}^{-}$ and B${}^{3+}$ resulted in different epitaxial growth rate. The primary nanoparticles attached along (001), (100) and (010) planes and formed octahedral-, diamond-like and plate-like particles. The as-prepared ZrO${}_{2}$ particles with different sizes and shapes exhibited different photocatalytic efficiency on degradation of organic dyes. Under UV irradiation, the highest 88% MB degradation was obtained within 60 min for ZrO${}_{2}$ photocatalyst synthesized with 0.01 mol/L NaBF${}_{4}$ mineralizer. These properties proved that the photocatalyst would be favorable for potential practical application. The future scope of this work includes the research of the growth mechanism with different morphologies ZrO${}_{2}$ particles more systematically, explore of its potential utilization as starting materials in preparing oriented polycrystalline zirconia ceramics and the evaluation of the stability of the resulting microstructure and performance.

## Figures and Tables

**Figure 1 nanomaterials-12-03405-f001:**
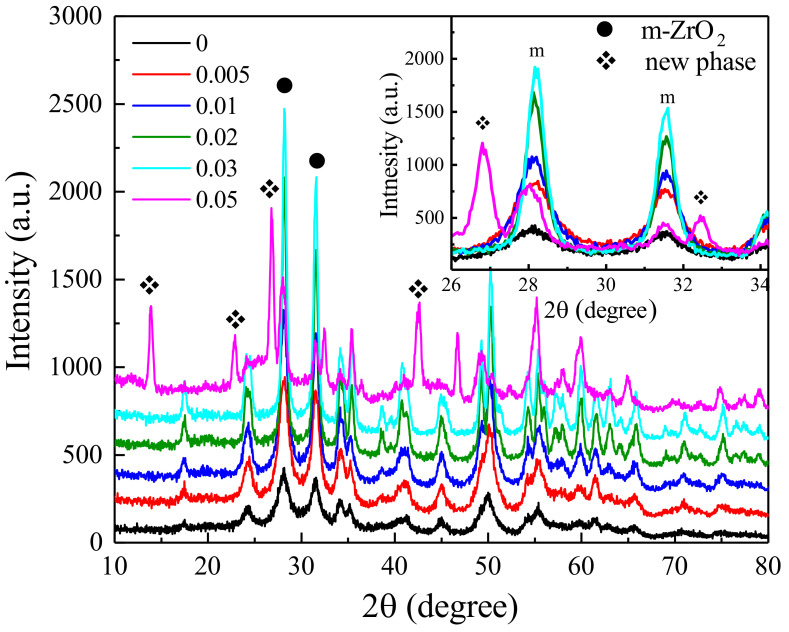
X-ray diffraction patterns of HT-ZrO${}_{2}$(*x*) synthesized with different concentration of NaBF${}_{4}$.

**Figure 2 nanomaterials-12-03405-f002:**
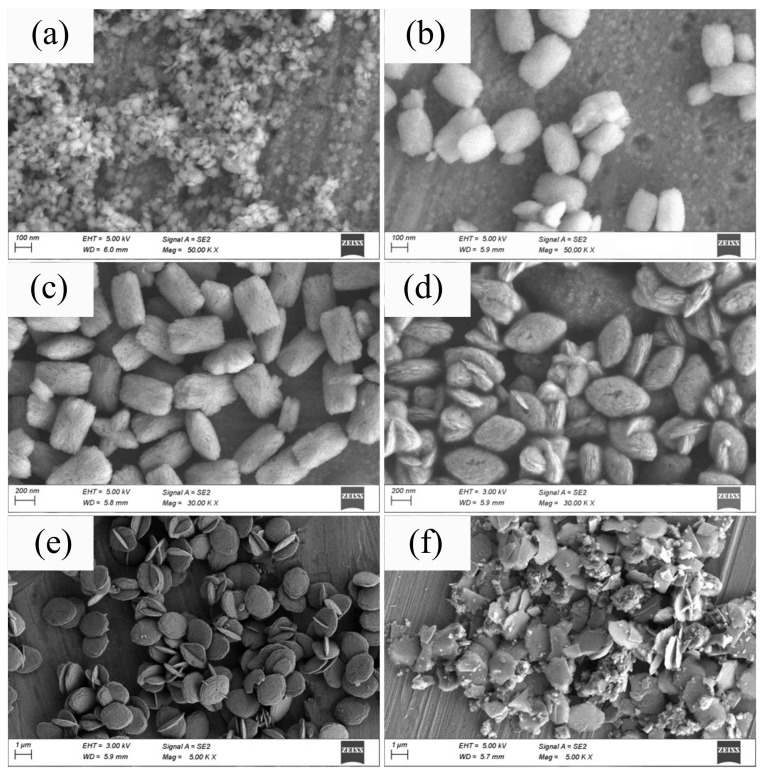
SEM micrographs of HT-ZrO${}_{2}$*x* particles synthesized with different concentration of NaBF${}_{4}$. (**a**) *x* = 0, (**b**) *x* = 0.005, (**c**) *x* = 0.01, (**d**) *x* = 0.02, (**e**) *x* = 0.03, (**f**) *x* = 0.05.

**Figure 3 nanomaterials-12-03405-f003:**
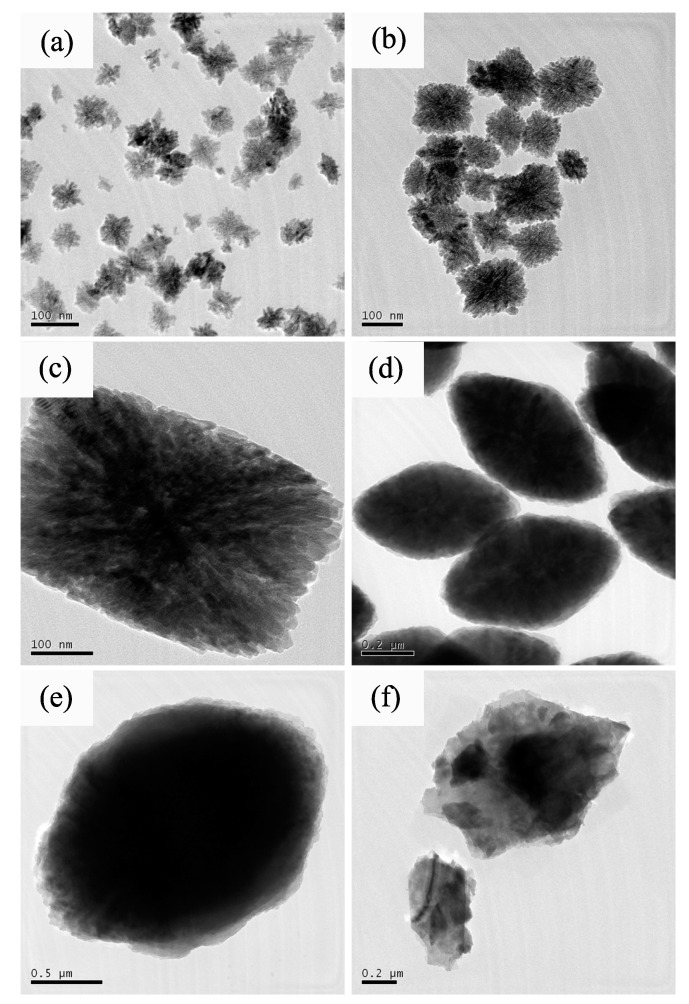
TEM micrographs of HT-ZrO${}_{2}$(x) particles synthesized with different concentration of NaBF${}_{4}$. (**a**) *x* = 0, (**b**) *x* = 0.005, (**c**) *x* = 0.01, (**d**) *x* = 0.02, (**e**) *x* = 0.03, (**f**) *x* = 0.05.

**Figure 4 nanomaterials-12-03405-f004:**
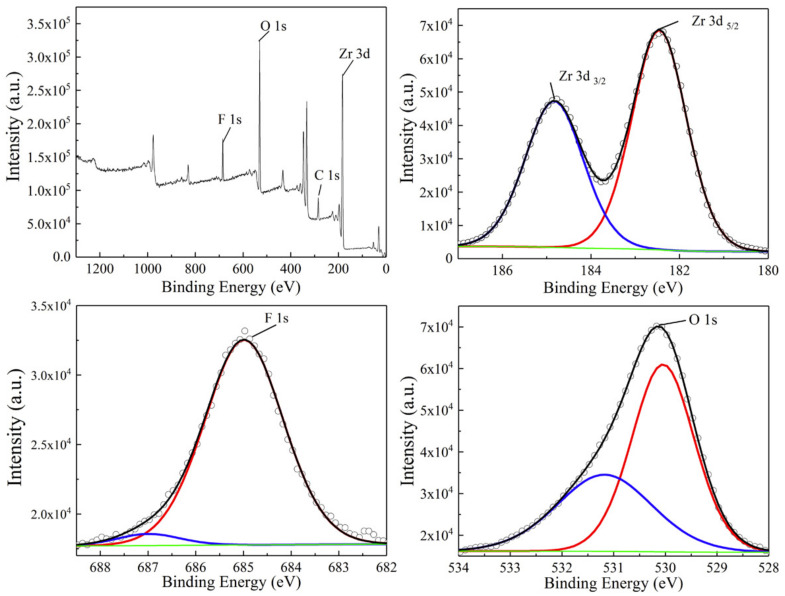
XPS survey and high resolution spectrums of HT-ZrO${}_{2}$(0.01) particles.

**Figure 5 nanomaterials-12-03405-f005:**
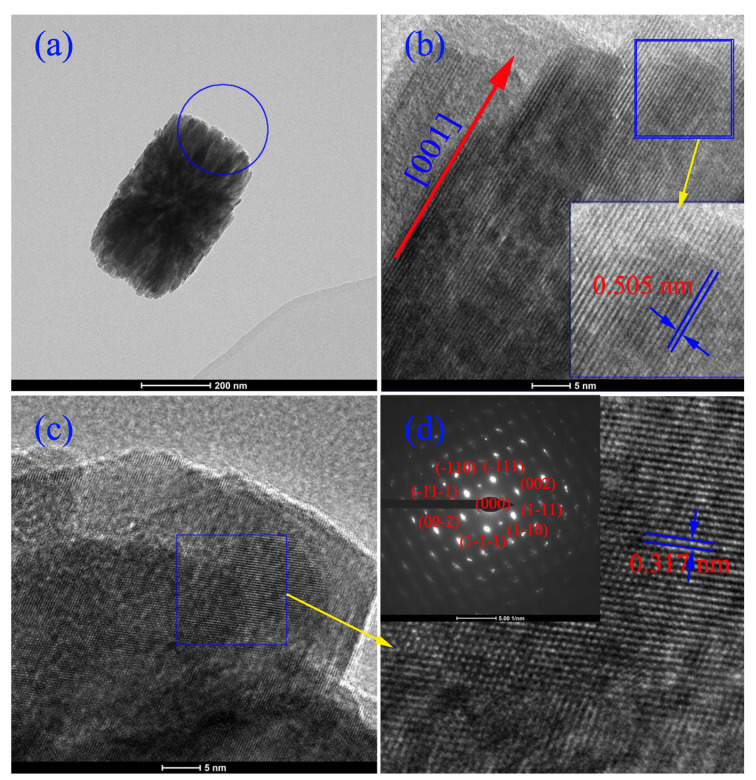
HRTEM and SEDA images of HT-ZrO${}_{2}$(0.01) particle with octahedron-like shape. (**a**) TEM image of a octahedron-like particle, (**b**) HRTEM image on [010] zone axis. (**c**) HRTRM image on [110] zone axis and its SAED pattern (**d**).

**Figure 6 nanomaterials-12-03405-f006:**
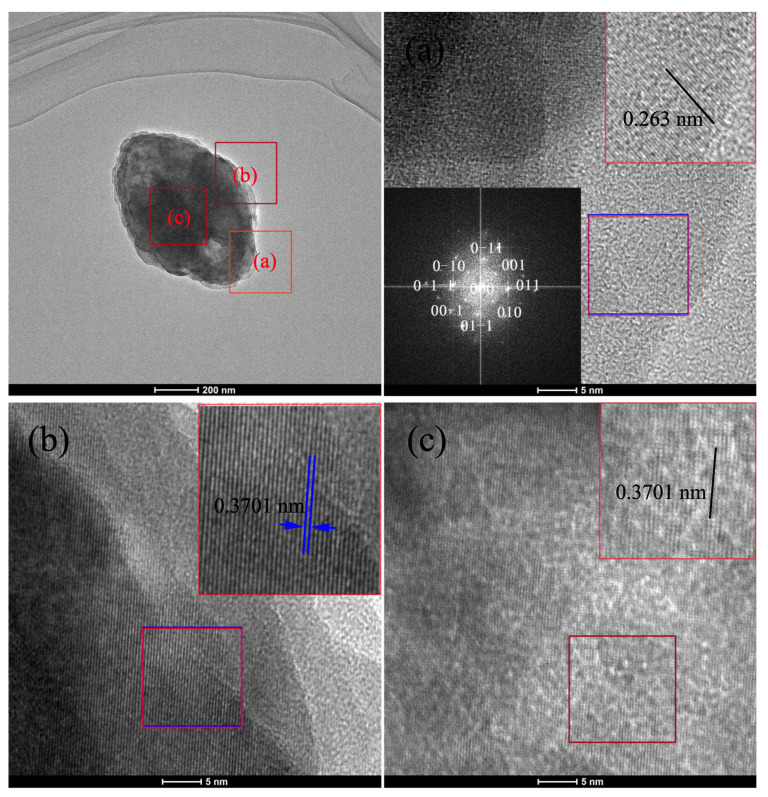
HRTEM and SEDA images of HT-ZrO${}_{2}$(0.02) particle with diamond-like shape. HRTEM images at the edge of the long axis (**a**), the edge of the short axis (**b**) and the particle center (**c**).

**Figure 7 nanomaterials-12-03405-f007:**
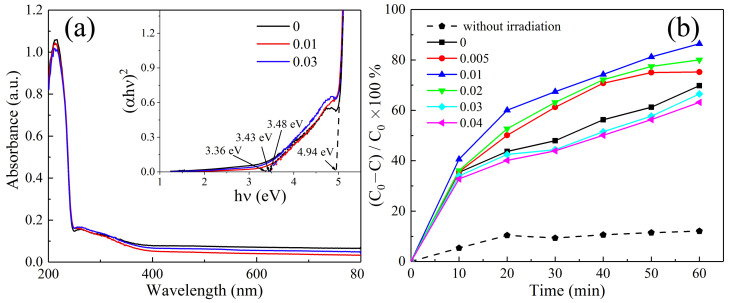
Physical properties of HT-ZrO${}_{2}$(*x*) particles (**a**) UV-Vis absorption spectrum, (**b**) photocatalytic degradation of MB.

**Table 1 nanomaterials-12-03405-t001:** Primary and secondary particle size of HT-ZrO${}_{2}$.

Concentration	Primary Crystallite Size (nm)	Secondary Particle Size			
		L (nm)	W (nm)	T (nm)	L:W
0	6.2	46.6	46.8	21.2	1:1
0.005	9.3	119.7	97.2	63.3	1.23:1
0.010	12.4	536.1	320.9	174.0	1.67:1
0.020	13.4	576.3	365.2	204.8	1.58:1
0.030	15.1	1995.0	1533.0	278.6	1.30:1

## Data Availability

The data in the article is valid.

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
