# Peer review of "Template-Free Hydrothermal Synthesis of Octahedron-, Diamond-, and Plate-like ZrO2 Mono-Dispersions"

_nanomaterials, 2022, doi:10.3390/nano12193405_

Round 1

Reviewer 1 Report

In general, the work is interesting, but without justifying the purpose of presenting possible practical applications, it looks rather average. I have doubts about the obtained results and their interpretation. I have presented my comments below:

1. In the introduction to the article, in the sentence concerning the use of nano- and submicron ZrO2 particles, reference should be made to literature or patents. I also believe that the practical application of materials based on ZrO2, depending on the morphology of crystallites, needs to be presented more broadly.

2. The introduction mainly describes the preparation and application of ZrO2 nanoparticles, which is confirmed by the quoted references, and the work itself is devoted to submicron crystallites. I believe that this should be taken into account as the formation of sub-micron structures may result from the agglomeration and growth of nanocrystallites. The selection of appropriate conditions for the hydrothermal reaction (time, temperature, and, of course, the use of surfactants, which was just underlined by the authors) could result in the formation of nanometric structures as a result of the synthesis.

3. p.2 line 68 - High revolution TEM (HRTEM) - should be "resolution"

4. p.2 line 80 - "UV-Cvis", I don't know this type of spectrophotometer.

5. p.3 line 91 and 92 "It is a pity that the diffraction peaks of secondary phase can not be indexed by any patterns based on the existing PDF data."  As for me, these are halogen or oxyhalide zirconium salts. It is also worth checking the databases, as they may be sodium or boron compounds enclosed in the structure of cystallites. However, this point should be clarified, for example, using crystallographic data processing programs (PANalythical X'Pert HighScore Plus, Match !, ACD Spectrum Processor). In the form in which the results of X-ray diffraction were presented in the article, one cannot speak of a monophasic structure of the material.

6. Due to the above, it would be highly recommended to include the results of material composition tests using ICP-OES or at least deep EDX (in different fields of material surface) methods. The presented results rather indicate a high content of admixtures, which may affect the shape of the crystallites and their properties, discussed in the article.

7. Zirconium halides are also Lewis acids, and admixture of this compounds may affect the structure and morphology of the obtained ZrO2 crystallites. That is why the presence of halogens in obtained material should be checked.

8. p. 5 line 214 various morpholoyies - various morphologies

9. For comparison, a figure showing the experiment with the addition of MB for the same samples, but without irradiation, must be provided separately, because the chemical sorption of MB on the surface and in the pores of the material can also cause the dye absorbance to disappear. Without the appropriate figures, the results presented are not reliable.

Author Response

We have read the reviewers’ comments very carefully and thank for their helpful comments. We have modified the manuscript accordingly, and the detailed corrections are listed below point by point:

Point 1: In the introduction to the article, in the sentence concerning the use of nano- and submicron ZrO2 particles, reference should be made to literature or patents. I also believe that the practical application of materials based on ZrO2, depending on the morphology of crystallites, needs to be presented more broadly.

 Response 1: The practical application of ZrO2 materials in the introduction part was mordified. The rang of application was presented more broadly, and the corresponging references were also added.

Point 2: The introduction mainly describes the preparation and application of ZrO2 nanoparticles, which is confirmed by the quoted references, and the work itself is devoted to submicron crystallites. I believe that this should be taken into account as the formation of sub-micron structures may result from the agglomeration and growth of nanocrystallites. The selection of appropriate conditions for the hydrothermal reaction (time, temperature, and, of course, the use of surfactants, which was just underlined by the authors) could result in the formation of nanometric structures as a result of the synthesis.

Response 2: The crystal growth mode and sub-micron particles can grow by aggregation mode with other nanoparticles are added into the introduction.

Point 3: p.2 line 68 - High revolution TEM (HRTEM) - should be "resolution"

Response 3: The miswritten word “revolution” has been corrected to “resolution”

Point 4: p.2 line 80 - "UV-Cvis", I don't know this type of spectrophotometer.

Response 4: The miswritten word “UV-Cvis” has been corrected to “UV-vis”.

Point 5: p.3 line 91 and 92 "It is a pity that the diffraction peaks of secondary phase can not be indexed by any patterns based on the existing PDF data." As for me, these are halogen or oxyhalide zirconium salts. It is also worth checking the databases, as they may be sodium or boron compounds enclosed in the structure of cystallites. However, this point should be clarified, for example, using crystallographic data processing programs (PANalythical X'Pert HighScore Plus, Match !, ACD Spectrum Processor). In the form in which the results of X-ray diffraction were presented in the article, one cannot speak of a monophasic structure of the material.

Response 5: The XRD results have been carefully compared with the standard PDF card. With our further study of the materials synthesized with this method on the condition with higer NaBF4 concentration, it is found that Na2ZrF6 was formed. Thouth the diffraction peaks do not fit the PDF card perfectly, we expect that the new phase should be a king of Zirconium fluoride salts. The detail explain of the new phase has been represented in the revised manucript.

Point 6: Due to the above, it would be highly recommended to include the results of material composition tests using ICP-OES or at least deep EDX (in different fields of material surface) methods. The presented results rather indicate a high content of admixtures, which may affect the shape of the crystallites and their properties, discussed in the article.

Response 6: Further experiment and EDS tests were charactered with the synthesized samples. It is found that the B3+ ions can not be tested by EDS, this result is the same as XPS results. The powders synthesized with higher NaBF4 concentration are composed with Na, Zr and F element , so that we respect the new phase is a kind of fluoro-zirconium compound. The detail information of the system has been discussed in the revised manucript.

Point 7: Zirconium halides are also Lewis acids, and admixture of this compounds may affect the structure and morphology of the obtained ZrO2 crystallites. That is why the presence of halogens in obtained material should be checked.

Response 7: The reviewer's comments have revealed the essence of the problem, and we have detailed described and supplemented the role of F- ion according to the comments.

Point 8: p. 5 line 214 various morpholoyies - various morphologies

Response 8: The miswritten word“morpholoyies” has been changed into “morphologies”

Point 9: For comparison, a figure showing the experiment with the addition of MB for the same samples, but without irradiation, must be provided separately, because the chemical sorption of MB on the surface and in the pores of the material can also cause the dye absorbance to disappear. Without the appropriate figures, the results presented are not reliable.

Response 9: The degradation curve of MB for the samples without irradition was added in figure 7.

Reviewer 2 Report

1.   What does the Epitaxial mean in the abstract of the manuscript?

2.   It seems that B3+ ions cannot exist in a hot aqueous solution. Among the hydrolysis products of NaBF4 must be boric acid. I think that the mention of free B3+ ions should be removed from the abstract of the manuscript.

3. In the current form of the manuscript, the experiment is very briefly described. It is not clear how many initial solution(s) are used in compare with reactor volume, what is the pressure reached in the hydrothermal reactor? Was it the same in all cases?  I think that the description of the experiment should be significantly expanded.

4.   What is pH of the initial solution(s) and what is pH after the completion of the reaction? What process takes place in the reactor, please write all possible chemical reaction(s)?

5.   It is possible that only by F- ions have the influence on the formation of ZrO2 nanoparticle. A similar effect has already been described in a number of closely related experimental (https://doi.org/10.1021/jp201292d ; https://doi.org/10.1016/j.ijhydene.2014.07.178 ) and review (https://doi.org/10.1016/j.ceramint.2021.11.028 ) articles.

6.   F- ions can be interact with the crystal lattice of ZrO2 nanoparticles, and not only adsorbed on the surface of ZrO2, as the authors claim. For example, https://doi.org/10.1016/j.jdent.2021.103772

7.  Indirect evidence of interaction in this complex system is the appearance of additional reflections in the powder diffraction pattern. I think that the structure of this compound must be described, at least its chemical composition must be identified. Perhaps this is the key point in understanding what is happening in the system.

Author Response

We have read the reviewers’ comments very carefully and thank for their helpful comments. We have modified the manuscript accordingly, and the detailed corrections are listed below point by point:

Point 1: What does the “Epitaxial” mean in the abstract of the manuscript?

Response 1: In 1998, Penn and Banfield presented a new crystal growth mechanism, the so-called "oriented attachment" mechanism, in which secondary mono-crystalline particles can be obtained through attachments of primary particles in an irreversible and highly oriented fashion. But a similar spontaneous self-assembly process on the deposition of Ag nanoparticles onto Cu substrates was founded by Averback et al, and they called "epitaxy". In order to avoid ambiguity, we changed the "epitaxial" to "Anistropic".

Point 2: It seems that B3+ions can not exist in a hot aqueous solution. Among the hydrolysis products of NaBF4 must be boric acid. I think that the mention of free B3+ ions should be removed from the abstract of the manuscript.

Response 2: EDS test of the prepared powder has been charactered on the powders, and found that B element did not appear in the crystal interior and surface. So that B3+ ions has been removed from the abstract of the manuscript.

Point 3: In the current form of the manuscript, the experiment is very briefly described. It is not clear how many initial solution(s) are used in compare with reactor volume, what is the pressure reached in the hydrothermal reactor? Was it the same in all cases? I think that the description of the experiment should be significantly expanded.

Response 3: The experimental process has been modified, and the specific process of the experiment was described in detail. Due to the pressure measurement could not be carried out with the hydrothermal kettle, the filling amount of initial solutions for all the samples were 60 % to ensure the consistency of the experimental process.

Point 4: What is pH of the initial solution(s) and what is pH after the completion of the reaction? What process takes place in the reactor, please write all possible chemical reaction(s)?

Response 4: The pH of the initial solution and the solution after reaction have been tested and were added in the manuscript. And the possible chemical reactions were allso added in the result and disscussion part.

Point 5: It is possible that only by F-ions have the influence on the formation of ZrO2 nanoparticle. A similar effect has already been described in a number of closely related experimental (https://doi.org/10.1021/jp201292d ; https://doi.org/10.1016/j.ijhydene.2014.07.178 ) and review (https://doi.org/10.1016/j.ceramint.2021.11.028) articles.

Response 5: We have carefully read the literatures provided by the reviewers,and find that these literatures are very helpful in understanding the role of F- in this study. We have cited these literatures in the manuscript.

Point 6: F- ions can be interact with the crystal lattice of ZrO2 nanoparticles, and not only adsorbed on the surface of ZrO2, as the authors claim. For example, https://doi.org/10.1016/j.jdent.2021.103772

Response 6: We have carefully read the literatures provided by the reviewers, and cited the literature in the manuscript.

Point 7: Indirect evidence of interaction in this complex system is the appearance of additional reflections in the powder diffraction pattern. I think that the structure of this compound must be described, at least its chemical composition must be identified. Perhaps this is the key point in understanding what is happening in the system.

Response 7: The XRD result of the new phase has been indexed with the standard PDF card, and the further experiment results with higher NaBF4 concentration inform the formation of Na2ZrF6, the detail explaination of the results were disscussed in the revised manucript.

Round 2

Reviewer 1 Report

I am satisfied with the responses presented by the authors and recommend this work for publication.

Author Response

We have read the reviewers’ comments and thank for their helpful comments.

Reviewer 2 Report

Thanks for the detailed comments.

Now the experiment can be reproduced by other scientists if necessary. And in the article a sufficient number of hypotheses about processes appeared.

Now I see only one typo on page 19 of the manuscript.  Under figure 1 there is one reference that can be removed (i hope). Please check this.

Author Response

We have read the reviewers’ comments very carefully and thank for their helpful comments. We have modified the manuscript accordingly, and the detailed corrections are listed below point by point: 

Point 1: Now I see only one typo on page 19 of the manuscript.  Under figure 1 there is one reference that can be removed (i hope). Please check this.

Response 1: The typographical error is due to the image being inserted into reference 32. and now  Fig. 1 has been reformatted on a new page.